Geographic range size and species morphology determines the organization of sponge host-guest interaction networks across tropical coral reefs

http://orcid.org/0000-0003-1449-6273 Pérez-Botello Antar Mijail 1 2 antarmijail@comunidad.unam.mx
http://orcid.org/0000-0002-4758-4379 Dáttilo Wesley 3
http://orcid.org/0000-0001-7490-3147 Simões Nuno 1 4 5 ns@ciencias.unam.mx
1 Unidad Multidisciplinaria de Docencia e Investigación, Facultad de Ciencias, Universidad Nacional Autónoma de México , Sisal, Yucatán , Mexico
2 Posgrado en Ciencias Biológicas, Universidad Nacional Autónoma de México , Ciudad de Mexico , Mexico
3 Red de Ecoetología, Instituto de Ecología A.C. , Xalapa, Veracruz , Mexico
4 Laboratorio Nacional de Resilencia Costera (LANRESC, CONACYT) , Sisal, Yucatan , Mexico
5 International Chair for Coastal and Marine Studies in Mexico, Harte Research Institute for Gulf of Mexico Studies, Texas A&M University, Corpus Christi, TX , United States of America
Venmathi Maran Balu Alagar
Electronic publication date: 2023 Nov 24
Publication date: 2023
Volume: 11
Electronic Location ID: e16381
Received 2023 Feb 6; Accepted 2023 Oct 9
Copyright: © 2023 Pérez Botello et al.
Copyright year: 2023
Copyright holder: Pérez Botello et al.
License: This is an open access article distributed under the terms of the Creative Commons Attribution License, which permits unrestricted use, distribution, reproduction and adaptation in any medium and for any purpose provided that it is properly attributed. For attribution, the original author(s), title, publication source (PeerJ) and either DOI or URL of the article must be cited.
License URL: https://creativecommons.org/licenses/by/4.0/

Keywords: Caribbean reefs, Community ecology, Marine ecology, Gulf of Mexico reefs, Functional traits, Ecology

Funding: Harte Research Institute Harte Charitable Foundation CONABIO-NE018, CONACyTCB-2012-01-177293 and PAPIIT IV300123 Unidad Multidisciplinaria de Docencia e Investigación—Sisal Facultad de Ciencias Universidad Nacional Autonomy de México (UMDI Sisal-FC-UNAM) CONACyT 2019-000037-02NACF This work was financed by grants to Nuno Simões by the Harte Research Institute, the Harte Charitable Foundation, CONABIO-NE018, CONACyTCB-2012-01-177293 and PAPIIT IV300123. This Project and Antar M Pérez-Botello was supported by the Unidad Multidisciplinaria de Docencia e Investigación—Sisal, Facultad de Ciencias, Universidad Nacional Autonomy de México (UMDI Sisal-FC-UNAM), and by CONACyT doctoral fellowship 2019-000037-02NACF through the PCB-FC-UNAM. There was no additional external funding received for this study. The funders had no role in study design, data collection and analysis, decision to publish, or preparation of the manuscript.

==============================
Sponges are widely spread organisms in the tropical reefs of the American Northwest-Atlantic Ocean, they structure ecosystems and provide services such as shelter, protection from predators, and food sources to a wide diversity of both vertebrates and invertebrates species. The high diversity of sponge-associated fauna can generate complex networks of species interactions over small and large spatial-temporal gradients. One way to start uncovering the organization of the sponge host-guest complex networks is to understand how the accumulated geographic area, the sponge morphology and, sponge taxonomy contributes to the connectivity of sponge species within such networks. This study is a meta-analysis based on previous sponge host-guest literature obtained in 65 scientific publications, yielding a total of 745 host-guest interactions between sponges and their associated fauna across the Caribbean Sea and the Gulf of Mexico. We analyzed the sponge species contribution to network organization in the Northwest Tropical Atlantic coral reefs by using the combination of seven complementary species-level descriptors and related this importance with three main traits, sponge-accumulated geographic area, functional sponge morphology, and sponges’ taxonomy bias. In general, we observed that sponges with a widespread distribution and a higher accumulated geographic area had a greater network structural contribution. Similarly, we also found that Cup-like and Massive functional morphologies trend to be shapes with a greater contribution to the interaction network organization compared to the Erect and Crust-like morphos. Lastly, we did not detect a taxonomy bias between interaction network organization and sponges’ orders. These results highlight the importance of a specific combination of sponge traits to promote the diversity of association between reef sponges and their guest species.

Introduction

The ecology of a species or community can be studied based on the functional characteristics of organisms (Gagic et al., 2015; Schmitz et al., 2015; Pimiento et al., 2020). A functional trait or characteristic is an organism’s distinguishable and quantifiable attribute, typically evaluated at the individual level and comparable across different species (Poff et al., 2006; Weiher, 2011). Classifying biodiversity according to some trait or functional group emphasizes the phenotypic differences between taxa and breaks the species ancestor-descendent phylogenetic link (McGill et al., 2006; Weiher, 2011; Gagic et al., 2015; Bagousse-Pinguet et al., 2019). Some functional traits such as feeding strategies, reproduction, dispersal, species distribution, response to environmental changes, species morphology or the ability of certain species to create microhabitats and vertical complexity, generally determine what, how, where and why we observe different species in an ecosystem (McGill et al., 2006; Weiher, 2011; Costello et al., 2015; Gagic et al., 2015; Beauchard et al., 2017; Pimiento et al., 2020).

Sponges, some of the oldest animals on Earth with fossil evidence dating back over 580 million years ago (Chen, 2012), are crucial components of modern marine environments, acting as keystone species in benthic habitats (Rütlzer, 2004; Wulff, 2006, 2016; Bell, 2008; González-Rivero, Yakob & Mumby, 2011; Maldonado et al., 2016; Brusca, Moore & Shuster, 2018). In particular, sponges can structure coral reef ecosystem performing multiple functions simultaneously, for example, sponge species are capable of filtering large volumes of water, enhancing primary production, participating in critical processes such as carbon, nitrogen, silicon, and oxygen cycles, and providing habitats for a wide variety of life forms which included organism with simple body plans (microorganisms) to complex ones (invertebrates and vertebrates) (Diaz & Rütlzer, 2001; Bell, 2008; Bell et al., 2013; Pawlik et al., 2013; Maldonado et al., 2016; Rossi et al., 2017). These previous citated studies have recognized the functional importance of sponges in coral reef ecosystems but this understanding has not always been translated into larger scale efforts (Bell, 2008; Gaüzère et al., 2022) and even less into no-trophic interaction networks ecology (i.e., host-guest interactions).

Host-guest interactions, refer to the co-occurrence of two different species in both space and time (Watson & Pollack, 1999; Baeza, 2015; Overstreet & Lotz, 2016). The key attribute of these associations is the use of one organism as habitat by another organism of a different species (Baeza, 2015; Overstreet & Lotz, 2016). In this context, when we know the costs and benefits inherent to these relationships, it is possible to classify the host-guest interactions into parasitism, mutualism, or commensalism (Thiel, 1999; Watson & Pollack, 1999; Thiel & Baeza, 2001; Baeza, 2015). While it is well-documented that different sponge species can be habitat facilitators for a wide range of organisms, the specific costs acquired by the sponge species involved in these interactions remain poorly understood (Duffy, 1992; Wulff, 1997; Bell, 2008; Maldonado et al., 2016). In order to maintain consistency in the interactions classification, the term “host-guest interaction” is assumed by this study to describe the relationship between sponges and the fauna that inhabit them.

Coral reefs are a dynamic and complex marine ecosystem involving a wide array of interacting organisms (Knowlton, 2001; Bauer, 2004; Hagedorn et al., 2015). The mutualistic relationship between cyanobacteria (Symbiodinium sp.) and scleractinian corals is crucial for the existence of this ecosystem, yet there are many other interspecific relationships that contribute to its complexity (Bauer, 2004; Hagedorn et al., 2015). One example is the host-guest relationship between sponges and the organisms that inhabit them (Diaz & Rütlzer, 2001; Hooper & van Soest, 2002b; Maldonado et al., 2016; Pérez-Botello & Simões, 2021). Sponges of class Demospongiae possess the highest number of host-guest associations diversity in coral reefs across the globe, highlighting Agelas, Aplysina, Xestospongia, and Callyspongia as the main hostesses genera (Maldonado et al., 2016). However, the study of ecological interactions in reef environments and even more in sponge-associated fauna focuses on analyzing interactions between paired species or on a small subset of networks.

In the Northwest Tropical Atlantic (NWTA) coral reefs (the Caribbean Sea, the Gulf of México, and Bermuda), sponges are a crucial benthic component along with scleractinian corals and macroalgae (González-Rivero, Yakob & Mumby, 2011). Phylum Porifera has a heterogeneous distribution, being Caribbean Sea reefs the ones with the highest species richness, then the Gulf of Mexico reefs, and finally Bermuda reefs (Ocean Biodiversity Information System (OBIS), 2020). These sponges could provide a series of services to several guest organisms; for example, sponges as habitat facilitators, can be a food source to the associated organisms, or can provide a certain degree of protection to the organisms that associate with them, either through the direct use of different structures as shelter or camouflage, or through the indirect protection derived from secondary metabolites produced by the sponge (Dembowska, 1926; McLay, 1983; Bedini, Canali & Bedini, 2003; Pawlik, 2011; Cruz Ferrer, 2014; Maldonado et al., 2016; Harada, Hayashi & Kagaya, 2020). Due to the multifunctionality that sponges present in the NWTA reefs, the community structure of sponge species directly influences other reef organisms (Bell, 2008). The NWTA sponge-associated fauna is incredibly diverse, with over 284 known associated species inhabiting 101 sponge species (Pérez-Botello & Simões, 2021). The main pattern is that host-guest associations are species-specific at sponge phylum level, meaning that sponges have the potential to maintain part of the biodiversity of reef systems (Maldonado et al., 2016; Pérez-Botello & Simões, 2021). However, when evaluating the sponge-guest species richness, a few sponge species, such as Ircinia strobilina (Lamarck, 1816), Callyspongia (Cladochalina) aculeata (Linnaeus, 1759), and I. felix (Duchassaing & Michelotti, 1864), concentrate 187 associated guest fauna (Dardeau, 1981; Carrera-Parra & Vargas-Hernández, 1997; Pérez-Botello, 2019), more than half of the associated species recorded, while 33 sponge species only interact with 51 spp., hosting between one to three spp. (for an exhaustive list of sponge-associated species diversity review Pérez-Botello & Simões (2021) and visit the website https://marinespeciesinteractions.org/?p=2302). These host-guest species interactions array generates a complex network where sponges act both as hosts and connectors within the network, emerging the question of what functional characteristics regulate the contribution that each sponge species has to the structure of the host-guest interactions network within the NWTA coral reefs?

In the present study, we evaluated three traits that could regulate the structure of the interactions network. First, the sponge-accumulated geographic area, or the habitat availability to be colonized, could regulate the contribution that each sponge species has to the structure of the network. In addition to the widely recognized ecological species-area pattern described by the Island Biogeography Theory (MacArthur & Wilson, 1963; Cornell & Lawton, 1992; Lawton, 1999; Losos et al., 2009), Galiana et al. (2018) proposed a relationship between geographic area and the probability of establishing a species interaction. We tested the hypotheses that there is a positive correlation between the sponge-accumulated geographic area and the network structural importance of each sponge species (Moulatlet, Dáttilo & Villalobos, 2023). The second tested trait is how habitat heterogeneity, measured as the functional sponge morphology, promotes the establishment of guest species. For example, sponges with large volumes could host a certain types of guest organisms than sponges with smaller volumes and more tightly packed shapes (Koukouras et al., 1996; Hooper & van Soest, 2002a; Maldonado et al., 2016; Pérez-Botello, 2019). For this reason, we also tested the hypotheses that there is a plausible suitable functional morphological group that promotes the establishment of sponge-guest interactions. Lastly, we tested if the sponges’ taxonomy bias could affect the importance of each taxon in the interaction network structure. In other guest-host interaction models, particularly mutualism between sea anemones and crustaceans of the Caribbean Sea, it has been demonstrated a pairwise taxonomic relationship between anemones and shrimps (Mascaró et al., 2012; McKeon & O’Donnell, 2015; Kou et al., 2015; Pérez-Botello, Mascaró & Simões, 2021), such as Bartholomea annulata (Le Sueur, 1817) and Condylactis gigantea (Weinland, 1860), two sea anemones of the Actiniaria order that concentrate the major diversity of anemone-shrimps as Alpheus armatus Rathbun, 1901, Ancylomenes pedersoni (Chace, 1958), Thor amboinensis (De Man, 1888) and Periclimenes yucatanicus (Ives, 1891) (Silbiger & Childress, 2008; McCammon, Sikkel & Nemeth, 2010; McCammon & Brooks, 2014). Therefore, there may be a pattern in which sponge orders with a higher associated species richness are essential to maintain the structure of the host-guest interaction network on reef sponges.

Ecological networks can be visualized as a series of interconnected nodes and edges; these nodes represent different species within the ecosystem, and the edges represent the relationships and interactions between them (Ramírez-Flores et al., 2015; Cantor et al., 2018; Dehling, 2018; Martínez-Falcón, Martínez-Adriano & Dáttilo, 2019). These fixed representations of ecological processes help to identify keystone species that maintain and connect the network and predict changes in ecological communities, providing a comprehensive understanding of the dynamics of biological interactions (Bastolla et al., 2009; Cantor et al., 2018). Applying a network analysis to infer functional roles has been previously employed in other ecological systems, such as seed dispersal (Vidal et al., 2014), insect-plant mutualistic networks (Bastolla et al., 2009), vertebrate scavenging (Sebastián-González et al., 2021), and plant-pollinator—protective ant—seed disperser multi-interaction networks (Dáttilo et al., 2016).

In this study we perform a large-scale community-level network analysis on sponge-associated fauna, offering a valuable insight into the ecological importance of coral reef sponges host-guest interactions. We aim to evaluate the structural importance of each sponge species and identify the keystone functions that maintain the organization of sponges host-guest interaction networks across NWTA coral reefs.

Materials and Methods

Database

We employed a bibliographic Database containing 101 host sponges species that inhabit the NWTA coral reefs, which were classified into 12 orders of class Demospongiae (Pérez-Botello & Simões, 2021; https://zenodo.org/record/3333023). In order to ensure the highest possible taxonomic resolution, the original database was filtered to include only sponge taxonomic entities at species level. The dataset analyzed in this meta-analysis comprised 65 scientific publications, yielding a total of 745 host-guest interactions between sponges and their associated fauna within the NWTA coral reefs (Dataset S1).

We employed the Marine Ecoregions of the World (MEOW) to systematically organize the host-guest interactions dataset in a cohesive manner (Spalding et al., 2007). The analyzed ecoregions were Bahamian, Bermuda, Eastern Caribbean, Floridian, Greater Antilles, Northern Gulf of Mexico, Southern Caribbean, Southern Gulf of Mexico, Southwestern Caribbean and Western Caribbean. The number of interactions, locations and publications records per ecoregion varied, with Bahamian recording 46 interactions across 28 locations and 14 publications, Bermuda recording eight interactions across one location and one publications, Eastern Caribbean recording 55 interactions across 17 locations and six publications, Floridian recording 46 interactions across 22 locations and 14 publications, Greater Antilles recording 66 interactions across 23 locations and eight publications, Southern Caribbean recording 106 interactions across 14 locations and 10 publications, Northern Gulf of Mexico recording 131 interactions across seven locations and nine publications, Southwestern Caribbean recording 77 interactions across 19 locations and 146 publications, Southern Gulf of Mexico recording 317 interactions across 19 locations and 10 publications, and Western Caribbean recording 110 interactions across 15 locations and 18 publications. A total of 76 sponge species were recorded hosting 268 sponge-associated fauna (Fig. 1).

Figure 1 Regionalized map of sponge host-guest interactions in the Northwestern Tropical Atlantic coral reefs.

The ecoregions are based on the Marine Ecoregions of the World classification (Spalding et al., 2007). Each region is labeled with the number of articles that provide information (r) and the number of recorded interactions (i). The sizes of each circle represent the number of citations that has each record. The colors of the circles represent the particular regionalization, gray for the Bahamian, light green for Bermuda, Blue for the Eastern Caribbean, light yellow for Floridian, dark yellow for the Grater Antilles, orange for the Northern Gulf of Mexico, sapphire for the Southeastern Caribbean, brown for the Southeastern Gulf of Mexico, light blue for the Southwestern Caribbean and dark green for the Western Caribbean. To interactively explore this map, visit the website https://marinespeciesinteractions.org/?p=2302.

To assess publication bias and measure the effect that could have in terms of replication we relate the number of publications by ecoregion (observed publication sample sizes) with the publications total number in the dataset (expected publication sample size) (Egger et al., 1997; Thompson, Smith & Sharp, 1997; Thompson & Sharp, 1999). With this citation proportion, we estimate the publication heterogeneity and evaluate the discrepancy between the observed and expected publication effort across ecoregions. The meta-bias analysis demonstrated that each ecoregion exhibited homogeneity in publications proportion (t2 estimator = Maximum-likelihood, p-value = 0.99), and no statistically significant publication bias was found between regions (bias method = Thompson, p-value = 0.28) (Fig. S1) indicating that the replication of the study across the different regions is reliable. We evaluated the publication bias using the functions ‘metaprop’ and ‘metabias’ from the package ‘meta’ in R (Schwarzer, Carpenter & Rücker, 2015; Balduzzi, Rücker & Schwarzer, 2019).

Interactive figures are available on Marine Species Interactions web site (https://marinespeciesinteractions.org/?p=2271) and the working datasets, are available on Zenodo (https://doi.org/10.5281/zenodo.7549399).

Characterizing the structure of sponge-assosiated fauna network

To analyze the sponge host-guest interactions, we employed a complex network approach (Dáttilo & Rico-Gray, 2018). The relationships (links) between different species are represented as interconnected nodes. Because the information on sponge-guest relationships used in this study came from different publication records employing several sampling methodologies and criteria for accounting interaction frequency, we used a qualitative network (Bellotti, 2014) in which a value of one represented the presence of an interaction between a sponge and one guest species, and 0 indicated the absence of a recorded interaction. This approach allowed us to analyze the contribution to network organization and species’ importance without the influence of changing sampling methods and, therefore, ensuring that all species report the same type of biological information (Bellotti, 2014; Dáttilo et al., 2016).

We begin by characterizing the contribution to network organization by using two structural properties frequently reported in species interaction networks: nestedness and modularity. In nested networks, species engaged in few interactions (i.e., potentially specialists) are connected to a subset of species engaged in many interactions (i.e., potentially generalists), while modularity describes a pattern where there are subgroups of species of one lower level (e.g., sponges) that interact strongly with a subgroup of species of another higher level (e.g., guests). Nestedness was estimated using the Nestedness Metric based on Overlap and Decreasing Fill (NODF; Almeida-Neto, Frensel & Ulrich, 2012), a nestedness descriptor that varies from zero (not nested) to 100 (perfectly nested). We estimated the Modularity (Q) using the QuanBiMo algorithm, which repeatedly divides a network into modules (we set to 107 swaps) and re-calculates modularity until reaching an optimal Q value, which ranges from zero (no more links within modules than expected by chance) to one (maximum possible modularity). Then, we generated 1,000 random matrices to test the significance of nestedness and modularity according to a null model, in which the number of interactions and the number of links (and hence connectance) keep constant. We calculated nestedness and modularity using the function ‘network level’ from the package ‘bipartite’ in R (Dormann, Gruber & Fründ, 2008).

Network structural contribution

We considered seven complementary species-level descriptors to measure the species’ contribution to network organization: species degree, betweenness, closeness, Katz centrality, among-module connectivity (ci), standardized within-module degree (zi), and contribution to nestedness (cni). We chose these descriptors because they provide complementary biological information on the contribution to network structure and, therefore, are expected to be more robust than on a single measure (Vidal et al., 2014; Corro et al., 2022). Degree centrality is the number of interactions established by a species (Borrett, 2012). Betweenness centrality calculates the fraction of the smallest number of links between any two species in a network that pass-through a given species. Closeness centrality is a measure of the average of the geodesic distances (shortest path lengths) from a focal species to all other species in the network. Biologically, a species with high closeness centrality is considered to be centrally located and has quick access to resources effectively throughout the network. Katz centrality calculates the number of immediate neighbors and the direct and indirect paths of a species to other species in the network (Katz, 1953). Among-module connectivity (ci) describes how evenly distributed are the interactions of a given species across modules (Olesen et al., 2007). Standardized within-module degree (zi) calculates of the extent to which each species is connected to the other species in its module within the network (Olesen et al., 2007). Contribution to nestedness (cni) is the degree to which the interaction of species increases or decreases the network’s overall nestedness (Saavedra et al., 2011). Because all these descriptors were highly correlated (Fig. S2) we used a principal component ordination (PCO) to reduce the seven-dimensional space to a single generalized index summarizing species’ contribution to network organization as often used in specialized literature. The first principal component (PCO1) accounted for 99.8% of the variability, indicating complementarity among the seven descriptors, where species with higher PCO1 scores exhibit many interactions, are connected to other species by multiple direct and indirect pathways, and represent a higher contribution to network organization. Since the values obtained from PCO1 are both positive and negative and the magnitude of change exceeds the tens between the minimum and maximum value, we changed the starting point of the values to zero, and we did a square root transformation of PCO1 (PCO11/2).

Sponge functional traits

Information on two functional traits that may influence a sponges’ ability to host a greater or lesser number of species was compiled: (1) The accumulated geographic area of host sponges and (2) host sponges’ functional morphology. The accumulated geographic area of sponges was determined using records from the Global Biodiversity Information Facility (GBIF, 2021), the interaction records of Pérez-Botello & Simões (2021) (Dataset S2), and the Caribbean and Gulf of Mexico coral reefs shapefile from the Allen Coral Atlas (Allen Coral Atlas, 2022). A circular buffer area of 10 m2 was calculated for each sponge record, and the area was cropped with the reefs’ shapefile. The results in square kilometers (km2) were used as colonizable sponges’ areas. Sponges functional morphology (Cust-like, Massive, Cup-like, and Erect; Fig. 2) were classified with the Schönberg (2021) morphological standardization (Dataset S3), and with the original descriptions of each sponge species when is available or with a species taxonomy update.

Figure 2 Illustrations of sponge’s functional morphologies.

Dark yellow for Crust-like, light blue for Massive, pink for Cup-like and orange for Erec. For more information on the morphological standardization used, please refer to Schönberg (2021).

Statistical analysis

To analyze the sponge contribution to the network concerning their geographic accumulated area, we used a generalized linear model (GLM: family = Poisson). This analysis provides a quantitative evaluation of the hypotheses that the sponges’ organization contribution to the interaction network is positively related by the accumulated geographic area of sponges. The sponges that provide a large geographical colonizable area potentially had greater structural importance than those sponge species with a restricted distribution (Dataset S4).

At the same time, we conducted an analysis of variance (ANOVA) to compare the different morphologies and test the hypotheses that sponges functional morphology improve the sponge contribution to the interaction network organization, i.e., some sponges morphologies are more suitable for hosting more guests than other morphologies (Dataset S4). We also evaluated the potential for sponges’ taxonomy bias in the network organization’s importance. To do so, we grouped the different sponge species by order and conducted another ANOVA to compare the taxonomic groups (Dataset S4).

Results

Network-level properties

The analysis of nonrandom patterns in the sponge-host interaction network of NWTA coral reefs demonstrated a statistically significant occurrence of nestedness and modularity in the organization of ecological network (NODF = 15.03 and M = 0.51, respectively. Both p-values < 0.001). A nested network exhibits a hierarchical organization of species interactions, specialist species engaged in few interactions are connected to a subset of generalist species with more interactions. Additionally, modularity describes a pattern where there are subgroups of guest species that interact more frequently with a group of host species. The presence of both nestedness and modularity network suggests a complex and dynamic structure within the sponge associated community, with host sponge species like Ircinia strobilina (PCO11/2 = 23.785; linked guest spp. 78) I. felix (PCO11/2 = 19.030; 46 guest spp.), Callyspongia aculeata (PCO11/2 = 18.121; 63 guest spp.), and C. fallax Duchassaing & Michelotti, 1864 (PCO11/2 = 16.389; 29 guest spp.) acting like species connectors and maintaining the network’s structure. At the same time this pattern exhibits a high diversity of low connected species, for example, Spongia (Spongia) obliqua Duchassaing & Michelotti, 1864 (PCO11/2 = 0.037; 1 guest sp.), Verongula rigida (Esper, 1794) (PCO11/2 = 0.088; 1 guest sp.), and Cribrochalina vasculum (Lamarck, 1814) (PCO11/2 = 1.004; 2 guest spp.). Despite their limited interactions, these low connected sponge species contribute to the overall sponge host diversity within the network (Fig. 3).

Figure 3 Sponge host-guest interaction network for the Northwestern Tropical Atlantic coral reefs.

The left side of the network represents the sponge-associated fauna, dark blue denotes Arthropods, red Annelids, green Vertebrates, golden Mollusks, pink Echinoderms and light-yellow Cnidarians. The left side of the network represents the host-sponges. The sponges are classified according to its functional morphology (Schönberg, 2021), pink for Cup-like, light blue for Massive, orange for Erect, and dark yellow for Crust-like. To interactively explore the network, visit the website https://marinespeciesinteractions.org/?p=2333.

The observed pattern highlights the remarkable dominance of interactions by a specific subset of sponge species within the ecological network. It is remarkable that a small number of sponge species play a crucial role in maintaining the organizational structure of the network through a high number of interactions, while a larger proportion of sponge species exhibit minimal associations with other species. In the same way, this pattern of a limited number of highly connected species making a significant contribution to the organization of the interaction network is constant regardless of the functional sponge morphology.

Trait-level properties

The sponge accumulated geographic area exhibited a positive relationship with the contribution of each sponge to the network organization (Intercept = 1.178, z value = 13.206, p-value < 0.001). Our findings reveal that as the sponge cumulative area increases, so does the contribution of sponges to network organization (Fig. 4). Widely distributed sponges such as Ircinia strobilina, I. felix or Callyspongia aculeata are essential species with a significant structural contribution to the network and a higher accumulated area. In contrast, sponges with focal distributions or sponges with only reported in unique sites such as Cliona vermifera Hancock, 1867, Aplysina bathyphila Maldonado & Young, 1998 Niphates erecta Duchassaing & Michelotti, 1864 and Ircinia ramosa (Keller, 1889) had a lower contribution to the network organization. In the same way sponges functional morphology also influenced the magnitude to which it contributed to the structure of the guest-host interaction network (ANOVA: F2.731 = 5.389; p-value < 0.002; Fig. 5). Sponges with Cup-like (mean ± s.e.: 8.348 ± 5.886) and Massive morphologies (7.844 ± 5.962) had a significant superior contribution to network organization than Erect (4.153 ± 2.625), and Crust-like (8.348 ± 1.746) morphologies. Finally, there were no significant differences in the contribution of sponge orders to the network organization (ANOVA: F1.941 = 0.587; p-value < 0.832; Fig. 6).

Figure 4 Relationship between sponge contribution to network organization and sponges accumulated geographic area.

General lineal model between sponge contribution to network organization and sponges accumulated geographic area (Intercept = 1.178, z value = 13.206, p-value < 0.001). The sponges are classified according to its functional morphology (Schönberg, 2021), pink for Cup-like, light blue for Massive, orange for Erect, and dark yellow for Crust-like.

Figure 5 Sponge contribution to network organization according to sponge functional morphology.

Crust-like and Massive morphologies are the statistically different groups (ANOVA: F2.731 = 5.389; p-value < 0.002). The sponges are classified according to its functional morphology, pink for Cup-like, light blue for Massive, orange for Erect, and dark yellow for Crust-like.

Figure 6 Sponge contribution to network organization according to sponge order.

No statical differences between sponge orders (ANOVA: F1.941 = 0.587; p-value < 0.832) are founded. The boxplot interquartile black line provides the mean value. Upper and lower whiskers extend to visual represent all data distribution.

Discussion

Our results provide evidence that supports the relationships between network organization, accumulated geographic area, and functional morphology in the interactions between sponge hosts and their associated guest species. The relationship between accumulated geographic area and network organization was already supported by the existing interactions theory (Galiana et al., 2018; Dallas & Jordano, 2021), however this result improves further theoretical development in coral reef interactions ecology. In contrast, although there are numerous studies of the actual fauna that use sponges as microhabitats (Bell, 2008; Maldonado et al., 2016; Pérez-Botello & Simões, 2021), there has been not clarify the sponge morphological features that make them a suitable environment.

The positive relationship between sponges’ contribution to network organization and its accumulated geographic area has a significant ecological implication. Our results indicate that as the accumulated geographic area increases so does the possibility to interact with a wide diverse guest species. For example, Ircinia strobilina, I. felix or Callyspongia aculeata are sponges with a wide distribution, also are generalist sponges in terms of hosted gest species. Isolated, each one of these three sponges can host up to 46 guest species of the five registered Phyla, like Arthropoda: Synalpheus townsendi Coutière, 1909, S. fritzmuelleri Coutière, 1909 and Colomastix heardi LeCroy, 1995; Annelida: Haplosyllis spongicola (Grube, 1855) and Loimia medusa (Savigny, 1822); Chordata: Elacatinus xanthiprora (Böhlke & Robins, 1968); Mollusca: Isognomon bicolor (C. B. Adams, 1845); Echinodermata: Ophiothrix (Ophiothrix) oerstedii Lütken, 1856; Cnidaria: Umimayanthus parasiticus (Duchassaing de Fonbressin & Michelotti, 1860); and together these sponge concentrates 187 guest species of the 745 recorded host guest-interactions (25% of the recorded interactions). According to this relationship sponges with less geographic accumulated area like, Aplysina bathyphila, Niphates erecta and I. ramosa had a lower contribution to the network organization, interacting with fewer guest species and showing a tendency to associate with guests like, S. townsendi and H. spongicola that are capable of colonizing a diverse array of sponge species. This relationship creates the situation where a sponge host already present in the network can increase the probability of interacting with a potential guest species if this sponge increases their geographical distribution, leading to a cascade effect between sponges’ availability and potential guest diversity in tropical Atlantic coral reefs. However, it is important to note that there is a group of sponges with a low accumulated area and few occurrence records, but their geographical distribution covers both the Caribbean and the Gulf of Mexico reefs. For example, Agelas dispar Duchassaing & Michelotti, 1864, Aplysina lacunosa (Lamarck, 1814), Ircinia campana (Lamarck, 1814) and Dragmacidon lunaecharta (Ridley & Dendy, 1886). This feature could indicate that these sponge species are difficult to detect or identify to species level in the field or that they have low abundances in the different reef systems they inhabit. A variable that could be explored in the future is the paired-wise distance between records of sponges of the same species and analyzing if the sum of these distances is related to the structural importance of each sponge species.

In the same way, sponges provide a complex living space for a large number of species from many taxa (Renard et al., 2013; Pérez-Botello & Simões, 2021). Most sponge associated fauna live inside the sponge whether using the canals, oscula, or pores of the sponges (Westinga & Hoetjes, 1981; Koukouras et al., 1996; Hultgren & Duffy, 2010). This fact generates that the interactions between sponges host and their guest species can be influenced by a particular sponges morphological features. Our results indicate that certain sponge functional morphotypes have an effect on the organization of the host-guest interaction networks. Specifically, we found that Cup-like and Massive functional morphologies, frequently are more important in the network organization than Erect and Crust-like functional morphotypes. Cup-like sponges have concave upper surface and can efficiently separate their in- and exhalant openings (Renard et al., 2013; Schönberg, 2021). This functional morphology commonly has a roughly cylindrical (tubes and barrels) or inverted-cone symmetry (cups) with a larger internal volume and wide tube oscula (entrance) and, if multiple Cup-like structures aggregates in a group of a single individual sponge, the sponge heterogeneity enhances (Schönberg, 2021). In addition, massive sponges are very roughly as wide as high; in many cases, this functional morphology has a unified body mass comprised of fused subunits, resulting in the formation of interconnected small cavities and microcompartments (Schönberg, 2021) Also, massive sponges are characterized by its remarkable robustness, making them capable of providing a stable habitat compared to other functional morphologies (Bell & Barnes, 2000; Schönberg, 2021). In contrast with these two volumetric and tridimensional complex functional morphologies Crust-like shapes combine encrusting and creeping sponges, resulting in a low-profile body shape that extends parallel to the substrate; this morpho have a larger surface area compared to their height, and lack three-dimensional or vertical structures (Schönberg, 2021). On the other hand, erect sponges have a small attachment area and are positioned away from the substrate, they exhibit a predominantly vertical orientation, minimizing their horizontal surface areas (Schönberg, 2021). Their vertical orientation and small attachment area make them susceptible to fragmentation, detachment, and removal by strong flow, waves, or storm surges (Wulff, 1995; Schönberg, 2021). This process could generate an unstable and dynamic habitat, despite the high sponge survival rate after fragmentation (Wulff, 1985, 1995, 1997, 2006).

Additionally, the abundance and species richness of sponge associated fauna could be regulated by three main morphological features. Firstly, sponges total volume. The number of sponges-inhabiting taxa is logarithmically related to sponge volume; larger sponges provide more substrate and resources for the colonization of different species, resulting in higher sponge-associated diversity (Westinga & Hoetjes, 1981). Secondly, the internal volume and diameter of the sponge’s canals. The shape and size of the sponge inner canals can physically limit the size of organisms that can inhabit the sponge’s interior; sponges with narrow canals provide suitable refuge for smaller organisms, while larger organisms are constrained by the reduced size and volume of the canals (Koukouras et al., 1996; Hultgren & Duffy, 2010). Lastly, the morphological heterogeneity of the sponge, including number of tubes, sponge surface, sponge total volume and sponge area, creates a complex and heterogeneous habitat that facilitate the colonization of different guest species (Pérez-Botello, 2019). As a result, it is the combination of these morphological features that makes Cup-Like and Massive sponges more significant in network organization and have a more diverse community of associated guests compared to Crust and Erect sponges.

In parallel, at sponge species level, it is very likely that the physical restrictions of each sponge species are promoting or limiting the diversity of the host-guest interactions. For example, volumetric and complex sponges like Callyspongia aculeata, C. fallax and Ap. fistularis, can establish a host-guest interaction with both small guest like Leucothoe spinicarpa (Abildgaard, 1789), Colomastix irciniae LeCroy, 1995 or Ophiactis quinqueradia Ljungman, 1872, that usually lives associated to the sponge surface and pores (Carrera-Parra & Vargas-Hernández, 1997; Crowe & Thomas, 2002; Winfield & Ortiz, 2010) and larger guest like Synalpheus hemphilli Coutière, 1909, S. townsendi or Pagurus brevidactylus (Stimpson, 1859) typically founded in the sponges canals (Christoffersen, 1979; Dardeau, 1981, 1984; Carrera-Parra & Vargas-Hernández, 1997; Ugalde García, 2014). In contrast restricted and spaceless morphologies, like the encrusting sponges Petrosia (Petrosia) weinbergi van Soest, 1980 or Cliona celata Grant, 1826 could mechanically limited the size range of potential guest species inhabiting the internal sponge cavities.

Furthermore, sponge morphology and guest behavior could regulate the diversity of observed interactions. Sponges with more complex, convoluted tube systems, such as C. aculeata and Aiolochroia crassa (Hyatt, 1875), tend to host more guest species than massive sponges with interconnected channels. For example, sponge species such as Hyattella intestinalis (Lamarck, 1814) are colonized only by Synalpheus species. These Crustaceans are extremely territorial, and some have a eusocial behavior like S. regalis defending the host sponge against other organisms (Duffy, 1996; Duffy & Macdonald, 1999; Duffy, Morrison & Macdonald, 2002). Probably the territorial behavior of certain guest species pushing another potential guest out of the sponge. The effect of this behavior could be reduced in sponge species with a system of tubes that allow different degrees of segmentation of the internal sponge volume.

Finally, the taxonomic bias analysis revealed a no significant impact in the organization of the sponge host-guest interaction network. This result suggests that, at least at the sponges’ order level, the relationship between sponges’ order and host-guest network organization is primarily regulated by the sponge-accumulated geographic area and the functional sponge morphology and, to a minor degree, by evolutionary processes. This pattern implies that the sponges host-guest interactions depend not only on sponge taxonomic classification but also on the functional roles and specific adaptations each sponge-host and associated-gest species develops. While this analysis revealed no statistically significant relationship at sponges’ order level, it remains plausible that certain guest species may exhibit specific associations with particular sponges. For instance, the annelid H. aplysinicola Lattig & Martin, 2011 was observed to exclusively interact with sponges belonging to the Aplysina genera, being a specialized host-guest interaction.

Conclusions

One of the main findings of our study is the positive relationship between accumulated geographic area and network organization. These results provide valuable information to the theory of ecological interaction networks in a marine environment. This research further enhances our understanding of how geographical distribution influences the diversity and complexity of host-guest interactions. We further demonstrate that as the accumulated geographic area of sponge hosts increases, so does the potential for interactions with a wide range of guest species. This effect highlights the significance of sponge availability in promoting guest diversity in Tropical Atlantic coral reefs.

Also, our study demonstrates the crucial role of sponge functional morphologies in the organization of host-guest interaction networks. Cup-like and Massive morphologies exhibit greater importance in network organization compared to Crust-like and Erect morphotypes. Cup-like sponges, with their volumetric canals, and the potential to generate three-dimensional complexity, provide diverse living spaces. Massive sponges, characterized by their robust structures and interconnected small cavities, offer a stable habitat for guest species colonization. In contrast, Crust-like with their low-profile body shapes and erect sponges with vertical orientations, implies space limitations and potential instability due to fragmentation and detachment.

Importantly, our research demonstrates that taxonomic classification has not significantly influenced the organization of host-guest interaction networks. Instead, the ecological patterns observed in these networks are primarily shaped by accumulated geographic area, and functional morphology. This highlights the need to consider the functional traits and characteristics of sponge hosts and their associated guest species in ecological studies.

Finally with this research, we generate a comprehensive understanding of the regional structure generated in the network of host-guest interactions of reef sponges. Being able to detect emerging patterns and keystone species within that network. Sponges such as Ircinia strobilina, I. felix, Callyspongia aculeata, and C. fallax were crucial in maintaining the network organization. The potential extinction of one or several of these species, or even a slight decrease in their presence within NWTA coral reefs, could trigger cascading losses in abundance and diversity of their associated fauna.

Supplemental Information

Supplemental Information 1 Meta-bias analysis results.

This graph exhibits the number of citations per EcoRegion and their corresponding proportion. Neither Citation Heterogeneity nor Citation Bias are statistically significant. The ecoregions are based on the Marine Ecoregions of the World classification (Spalding et al., 2007)

Click here for additional data file.

Supplemental Information 2 Principal Component Ordination of the seven species-level descriptors.

Purple dots represents each sponge species, and the similarity between sponges (Euclidean distance) are calculated according nine network centrality index (species-level descriptors): species degree, betweenness, closeness, Katz centrality, among-module connectivity (C_value), standardized within-module degree (Z_value), and nestedness contribution. We use Pearson’s correlation for the species-level descriptors.

Click here for additional data file.

Supplemental Information 3 Sponge interactions records for the Northwest Tropical Atlantic coral reefs.

The information of the sponge-hosts and sponges-guests taxonomic classification. Each interaction record is associated with the following information: country, locality name, Spalding et al. (2007) EcoRegion, Geographical latitude and longitude, and a Bibliographic reference.

Click here for additional data file.

Supplemental Information 4 Sponges distribution records according to GBIF and Perez-Botello & Simões (2022).

Each sponge record is associated with the following information: geographical latitude and longitude and with the information source of that record.

Click here for additional data file.

Supplemental Information 5 Sponges functional morphological.

Sponges morphological description, classification and functional morphological standardization according to Schönberg (2021).

Click here for additional data file.

Supplemental Information 6 Network Structural Index and Sponge accumulated area.

Each sponge species is associated with their taxonomic classification, the seven-centrality index, first Principal Component, sponge accumulated geographic area in square kilometers and sponges functional morphology.

Click here for additional data file.

We gratefully acknowledges the Biological Science Program, UNAM (Posgrado en Ciencias de la Biológicas, Facultad de Ciencias, Universidad Nacional Autónoma de México; PCB-FC-UNAM). We thank Diana Ugalde (DU), an expert in sponge taxonomy, for her input on the morphological classification of the recorded sponges in this study. Also, to the entire Biodiversidad Marina de Yucatán (BDMY) team by the constant exchange of ideas which generates a fertile land for scientific creativity and innovation. Finally, we gratefully acknowledge the IABO Hub for hosting this article.

Additional Information and Declarations

Competing Interests

Author Contributions

Data Availability

The authors declare that they have no competing interests.

Antar Mijail Pérez-Botello conceived and designed the experiments, analyzed the data, prepared figures and/or tables, authored or reviewed drafts of the article, and approved the final draft.

Wesley Dáttilo conceived and designed the experiments, analyzed the data, authored or reviewed drafts of the article, and approved the final draft.

Nuno Simões conceived and designed the experiments, authored or reviewed drafts of the article, and approved the final draft.

The following information was supplied regarding data availability:

The data is available at Zenodo: Antar Mijail Pérez-Botello, Wesley Dattilo, & Nuno Simões. (2023). Geographic range size and species morphology determines the organization of sponge host-guest interaction networks across tropical coral reefs (Raw data) (3.0) [Data set]. Zenodo. https://doi.org/10.5281/zenodo.8171115.

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
