# Peer review of "Geographic range size and species morphology determines the organization of sponge host-guest interaction networks across tropical coral reefs"

_PeerJ, doi:10.7717/peerj.16381_

## Round 0.1 · original submission · Major Revisions

This study is relevant to understanding the structure of sponge communities and sponge host-guest interaction networks across tropical coral reefs. Three reviewers have given their comments and suggestions to improvise the manuscript. Reviewer 2 has given elaborate guidance on how to improvise this manuscript and pointed out that "the ecological context and the hypotheses to elucidate an overall pattern beyond what previous studies have found are not clearly stated". Hence, follow all reviewers' comments and suggestions carefully and revise the manuscript.

·

Basic reporting

- The authors should improve English grammar and usage.
- The authors should provide locations in Figure 1
- The authors should give clear hypotheses

Experimental design

- Provide clearer research questions
- The authors should provide details of the methods.

Validity of the findings

- Provide clearer conclusion

Additional comments

- This manuscript should be improved for better understanding of the readers.

Reviewer 2 ·

Basic reporting

This manuscript evaluates through a meta-analysis (= a statistical analysis that combines the results of multiple scientific studies) the structural importance of sponge species in hos-guest interactions. The hosts' structural importance is based on two functional traits, sponge species' accumulated geographical area and sponge morphology. The meta-analysis is based on a filtrated version of the bibliographic database by Pérez-Botello and Simões, 2021(https://zenodo.org/record/3333023) and includes records at the species level. An ecological network analysis, a Generalized Linear Model (GLM), a Principal Component Ordination (PCO), and an Analysis of Variance (ANOVA) were utilized to identify holistic properties within host-guest interactions. Overall this study is relevant to understanding how sponge communities structure and how much associated biodiversity they host. However, the ecological context and the hypotheses to elucidate an overall pattern beyond what previous studies have found are not clearly stated. In addition, the network analysis presentation disregards the hosts' functional traits and lacks ecological relevance. The assigned morphological characteristics of the species are not specified and could be biased. The PCO is not discriminated by functional traits (or factors), so observing grouping by sponge functional patterns is impossible. Overall, the discussion of the manuscript lacks the ecological context and background on sponge morphology (incorrect assignments for many species), sponge-host interactions and fails to support the conclusions. The discussion travels too far from the data. It does not focus on what the data prove or demonstrate how the obtained results expand previous knowledge or support and challenge previous paradigms. In addition, the manuscript's English language and writing style needs improvement.

Experimental design

The experimental design (how many studies and from which locations were finally used for this specific metanalysis?) is not specified. How many records of host species were assessed per location? What is the bias in terms of replication? Please include this information in the methods. Also, the tables in the Supplementary Section with associated fauna and records should be merged, and a column with the location for each study should be included.

Validity of the findings

Results

The PCO figure should be discriminated by functional traits (i.e., morphology). The network, patterns, and major contributors to the network structure are poorly explained. Overall, the results do not clarify how host-guest interactions vary based on functional groups; what are the major patterns? And how it varies across species.

Figure 1. This figure refers to all the records obtained by the network and is not specific to the records used in this study after filtration. I suggest correcting this figure and including only the records used in this study after the data was filtrated (many records come from the same authors, so that you could add numbers or color-code per author). Also, adding the replicates (n= the number of host-species interactions) per location can help to understand the contribution of the studies and the number of records per location.

Figure 2- This figure, as it is, lacks the most relevant information related to the functional characteristics of sponges. To explicitly show the functional characteristics of sponges and how they connect to specific hosts, sponges should be divided by functional characteristics such as 1. generalists and specialists, 2. accumulated geographic area, and 3. morphology of host sponges.

Figure 3. Add the species in colors or morphological characteristics to this figure to detect any pattern in species or morphology and in relation to accumulated geographic area.

Figure 4- The photograph selected to represent encrusting species is from an “excavating” sponge, not from an encrusting species, and the spherical representative species is a common massive species; as their growth patterns vary, please select a typically spherical species. Checking the supplementary list, I found several errors related to the morphology assigned to the species. I recommend you include/collaborate with a sponge expert taxonomist or ecologist to re-organize this data.


Discussion

Lines 284-287- Vague sentence “However, our findings have the potential to
provide new insights into predicting the functional responses of faunal assemblages
associated with sponges to potential species extinctions in endangered coral reef
ecosystems.” Please specifically state how the study results provide new insights into predicting the functional responses.

Lines 292-301-
This section in the discussion has interesting statements. However, it is vague and does not express why the results are important and how this knowledge contributes to the structure of interaction networks in sponge communities and associated fauna “Galiana et al. (2018) proposed a relationship between geographic area and the probability of establishing a species interaction. Especially they found that widespread species are more likely to interact with other species. Our results indicate that as the sponge's accumulated geographic area increases, so does the potential for interactions with other organisms” The authors mention that widespread sponge species (as sponge accumulated geographic area) tend to have more potential for interactions with other organisms. Still, it does not specify which species of sponges are those and with which organisms those interactions occur. Instead, keep generalizing “Widely distributed sponge species are found to be particularly important in shaping the structure of the interaction network".

Please explain what significance you found in your results. The following sentence is general and does not expose the relevance of the results or why it is important in the context of this study “The positive relationship between the importance of a sponge species within the contribution to network organization and its accumulated geographic area is crucial to understanding the significance of these widely distributed sponge species and the vulnerability of specific geographically restricted interactions.”

Lines 328-329- The authors focus on functional characteristics of the sponge they did not measure, such as the physical structure of the sponges (volume, oscula size, number of tubes), but their never organized sponge species based on these characteristics for their analyses. The only morphological characteristic is general morphology, which seems biased, as not all photographs match the correct morphological characteristics for the representative species (i.e., Figure 4).

Lines 324-327- How do you verify that organisms inside the internal space were not excluded in encrusting species? Also, the internal space was not quantified, measured, or even supported by referenced research for each species. Therefore this statement is invalid “This sponge network importance, biological, translates to the fact that morphos that promote spatial micro-compartmentalization could support a more diverse community of associated fauna than morphos with less internal space”. What do you define as micro-compartmentalization?

The limitations of this study are not clearly stated in the discussion. There is no clear context for the findings concerning previous literature and results.

Lines 343-366- If you want to discuss the relevance of sponge-host interactions concerning chemical defenses, please categorize the species in your metanalyses based on chemical defenses and then evaluate directly in the network analyses your findings. I encourage the authors to include this hypothesis in their network analysis, make it part of this research, focus the network analysis, and develop a more substantial discussion.

Additional comments

Formatting and Language

The English language needs improvement to communicate ideas better so an international audience can understand your text. For example, the first line of the abstract, "Sponges have widely spread organisms in the tropical reefs", should be rephrased; sponges are organisms; they do not have organisms. Also, "behaving like ecosystem engineers" should be replaced by ecological terminology such as "they structure ecosystems". In addition, throughout the text, statements are placed without connecting ideas. For example, in the first paragraph of the introduction, the following three statements are three disconnected sentences of definitions that lack connective words between them "The ecology of a species or community can be studied based on the functional characteristics of organisms (Gagic et al. 2015; Schmitz et al. 2015; Pimiento et al. 2020). A functional trait is an organism's distinguishable and quantifiable attribute, typically evaluated at the individual level and compared across different species (Poff et al. 2006; Weiher 2011). Classifying biodiversity according to some trait or functional group emphasizes the phenotypic differences between taxa and breaks the species ancestor-descendent phylogenetic link (McGill et al. 2006; Weiher 2011; Gagic et al. 2015; Bagousse-Pinguet et al. 2019)."
The connection between statements and ideas needs to be clarified throughout the document. Repetition of statements without explicitly mentioning how this study contributes and connects to previous work is found in the introduction and discussion.

Format- Several mistakes, including misspellings of species names “ Aplisyna“ and words Line 503 “Graunds”, grammar Line 59 “2020)..”, and scientific notation Line 222 “10m2”, are found throughout the text. Also, some references are missing pages.


Abstract
The abstract does not reflect the methodology of this study. This study is a meta-analysis, which should be stated in the abstract, specifying that the data is based on previous literature obtained xxx across the Caribbean Sea and the Gulf of Mexico.

Introduction

The study is based on a database with scientific papers about sponge and host interactions. However, the introduction needs a general background on sponges as hosts of fauna on Caribbean reefs. Only one reference is mentioned with a single statement “Sponges of the Demospongiae Class possess the highest number of guest-host associations diversity in coral reefs across the globe, highlighting Agelas, Aplisyna, Xestospongia, and Callyspongia as the main hostesses Genera (Maldonado et al. 2017).” This reference concerns sponge grounds, not Caribbean coral reef sponges. Also, the introduction lacks ecological literature on sponge-host interactions (specifically from the studies included in the meta-analysis). I suggest the authors write a paragraph on what we know about sponge host-guest interactions at local levels from previous studies on the Gulf of Mexico and Caribbean coral reefs. Explain the general patterns from previous research, and state how much a network analysis can, in this context, tell us more that we do not already know. This study lack of a hypothesis. What do authors expect this metanalysis and network analysis will clarify, show, and highlight? The paradigm or hypothesis that may corroborate or dispute should be stated.

Reviewer 3 ·

Basic reporting

• The English language should be improved to ensure that an international audience can clearly understand your text. English grammar and punctuations to be thoroughly checked.
• Spell checks to be done.
• Spacing and alignment issues in manuscript to be looked into.
• Introduction & background needs more elaboration, needs more detail.
• Literature well referenced & relevant.
• Discussion is not adequate, I felt.
• Discussion should be focusing more on the results obtained.
• Introduction portions are in Discussion, to be re framed.
• Figures are relevant, quality needs to be improved in some figures, however, I have suggested some changes in legend colouring in the map.

Experimental design

• Research question well defined, relevant & meaningful. However, authors should clearly mention the knowledge gap their research fills.
• Methodically explained the steps of conducting this research. The methodology looks good.
• The text needs to be more seamless. It feels very disjointed as if each section is written by a different co-author. There is a dissonance in the writing, making it a little difficult for the reader to quickly comprehend.

Validity of the findings

• Conclusions are linked to the original research question & are supporting results. However, it does not link well with the discussion. It is advised to rework the discussion part for writing a better conclusion.
• All underlying data have been provided.
• It would have made a stronger paper if there was also proper field collection collected and studied by the authors, to better support the dataset information. However, the outcome of this work seems to have potential in the future.

Additional comments

The manuscript PDF is annotated for authors to see and check.
The manuscript needs to be restructured essentially in the Introduction and discussion part to convey the scientific potential of this research. In this context, I am recommending the manuscript for a major revisions to incorporate the suggestions mentioned in the PDF version of the Manuscript, which is attached here before the acceptance.

Annotated reviews are not available for download in order to protect the identity of reviewers who chose to remain anonymous.

---

## Round 0.2 · Minor Revisions

The authors have done substantial revision but it needs some more improvement for this manuscript to be published, hence I advise minor revision as per the reviewer's comments.

·

Basic reporting

1. The language is clear and comprehensible.
2. Literature cited is recent. 2022 (7), 2021(31)
3. The sources of the meta data have been cited.
4. Article type: in silico analysis of secondary data followed by analysis of interaction.

Experimental design

The statistical methods which have been applied are standard.

Validity of the findings

The findings highlight the interaction between sponges and associated fauna. it highlights gaps in the current research and suggests future directions.

Additional comments

The authors should include a graphical representation of the different types of sponges in the introduction. Diverse morphologies have been described, but there is not photographic information.

---

## Round 0.3 · accepted · Accept

The authors have given responses to all the queries of the reviewers. Hence, I accept this manuscript after some minor revisions. However, I advise the authors to follow an important suggestion of the reviewer to include the graphical representations of the sponge morphology as described in the discussion.

·

Basic reporting

This is the second version. The changes have been made by the authors.

Experimental design

Refer previous comments. No further changes required.

Validity of the findings

Refer previous comments. No further changes required.

Additional comments

Refer previous comments. No further changes required.